# A Unified Multi-Task Learning Framework for Generative Auto-Bidding with Validation-Aligned Optimization

## Abstract

In online advertising, heterogeneous advertiser requirements give rise to numerous customized bidding tasks that are typically optimized independently, resulting in extensive computation and limited data efficiency. Multi-task learning offers a principled framework to train these tasks jointly through shared representations. However, existing multi-task optimization strategies are primarily guided by training dynamics and often generalize poorly in volatile bidding environments. To this end, we present Validation-Aligned Multi-task Optimization (VAMO), which adaptively assigns task weights based on the alignment between per-task training gradients and a held-out validation gradient, thereby steering updates toward validation improvement and better matching deployment objectives. We further equip the framework with a periodicity-aware temporal module and couple it with an advanced generative auto-bidding backbone to enhance cross-task transfer of seasonal structure and strengthen bidding performance. Meanwhile, we provide theoretical insights into the proposed method, e.g., convergence guarantee and alignment analysis. Extensive experiments on both simulated and large-scale real-world advertising systems consistently demonstrate significant improvements over typical baselines, illuminating the effectiveness of the proposed approach.

## 1 Introduction

In modern advertising platforms, auto-bidding tasks play a critical role in optimizing campaign performance (He et al., 2021; Mou et al., 2022; Guo et al., 2024). To accommodate advertisers' varying demands, advertising platforms offer a range of bidding campaign types (Borissov et al., 2010; Aggarwal et al., 2024; Li et al., 2025). Beyond the primary campaign types, there exist several less commonly used types, such as supplementary budgets added to the primary campaign to optimize metrics like overall store conversions, direct conversions, and cart additions. For lower-usage campaign types, training dedicated automated bidding models for each type often yields limited performance gains due to data scarcity and incurs substantial maintenance overhead (Wang et al., 2023), thereby motivating the development of multi-task learning (MTL) frameworks that share representations across related objectives (Wu et al., 2020; Zhang & Yang, 2021).

**Pitfalls of applying naive MTL to online auto-bidding.** Although MTL provides a promising framework for handling heterogeneous tasks (Navon et al., 2022; Zhang et al., 2022), directly applying it to online advertising poses substantial challenges due to the highly volatile and uncertain nature of bidding environments (Zhao et al., 2020; Gao et al., 2025), as shown in Fig. 1. In practice, user behavior evolves rapidly and competitor strategies adjust unpredictably, creating frequent distributional shifts that are often abrupt and difficult to anticipate (Qin et al., 2025). Such nonstationary dynamics not only compromise the reliability and stability of learned models but also result in suboptimal bidding decisions that directly translate into degraded deployment performance in live systems (Chen et al., 2018a; Gao et al., 2025). Importantly, the consequences of these distributional shifts are not uniform across tasks. Certain tasks might leverage the changing patterns to improve short-term effectiveness, whereas others are more vulnerable and tend to overfit to transient fluctuations, thereby weakening the model's overall generalization ability. All of these practical scenarios give rise to two central research questions: (i) How to design adaptive mechanisms that can robustly suppress or mitigate the negative impact of unpredictable online distribution shifts, and (ii)

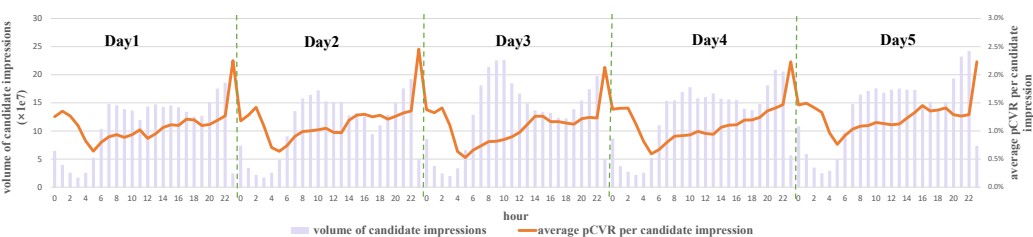

(a) Hourly volume of candidate impressions (left axis) and average value per candidate impression (right axis).

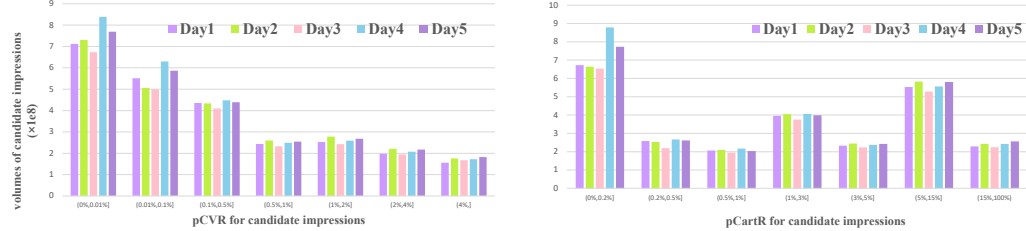

(b) Daily distribution of candidate impression values for two tasks, illustrating task-specific distribution shifts across days.

Figure 1: **Periodic patterns in nonstationary environments.** The bidding environments exhibit nonstationary dynamics with recurring temporal structures, such as diurnal periodicity.

How to construct a unified multi-task framework that not only facilitates effective knowledge transfer across related tasks, but also utilizes changing dynamics. This work will answer the above research questions in the context of the generative auto-bidding paradigm (Guo et al., 2024; Li et al., 2025).

**Generalization-aware task reweighting from online crafted distribution shifts.** For adaptive mechanism design, we propose *validation-aligned optimization*, which directly links task prioritization to validation performance in order to derive adaptive task weights. Over the course of online auto-bidding, the validation set is constructed by reserving full days from the training period without temporal overlap. This manner preserves structure and mimics the distribution shift between training and deployment. The task-specific weights are derived by aligning each task's training gradient with the gradient of the total validation loss, which serves as a reference for generalization improvement. Consequently, tasks with higher alignment receive larger weights, which emphasizes updates with stronger real-world effectiveness and aligns training dynamics with deployment objectives.

**Unified MTL structure for auto-bidding with temporal modules.** The design of the model architecture is equally critical for enabling effective knowledge transfer. Although each bidding task targets distinct objectives, they operate within a shared bidding environment, creating opportunities for joint learning. Despite the inherent nonstationarity of bidding dynamics, we observe some temporal patterns such as diurnal cycles that provide consistent signals in the environment, as shown in Fig. 1a. These recurring structures offer a valuable anchor for learning transferable representations. To exploit this property, we incorporate a dedicated temporal module to capture multi-scale periodicity in auction dynamics. By integrating this temporal module into the advanced generative auto-bidding paradigm (Guo et al., 2024; Li et al., 2025; Gao et al., 2025), we develop a unified multi-task learning framework that effectively fulfills multiple auto-bidding tasks.

To summarize, our contributions are three-fold:

1. We propose Validation-Aligned Multi-task Optimization (VAMO) that uses validation gradient feedback to guide training updates toward better generalization;

2. We design a unified multi-task learning framework that is built upon the emerging generative auto-bidding paradigm, incorporating a dedicated temporal modeling to capture periodic auction dynamics and enhance cross-task knowledge transfer;

3. We provide theoretical justification through convergence guarantee and alignment analysis linked to our strategy VAMO.

Extensive simulated and real-world experiments demonstrate significant performance improvements, offering practical insights for industrial deployment.

## 2 PRELIMINARIES

Auto-bidding seeks a bidding policy that maximizes the cumulative value of impressions won over a finite bidding episode, e.g., one day (He et al., 2021; Mou et al., 2022). Formally, the auto-bidding problem is usually modeled as a Markov Decision Process (MDP) defined by the tuple $< \mathcal{S}, \mathcal{A}, \mathcal{R}, \mathcal{P} >$. At each discrete time step $t \in [T]$, the state $\boldsymbol{s}_t \in \mathcal{S}$ describes the real-time advertising status that includes the remaining time, left budget, consumption speed, etc. The action $\boldsymbol{a}_t \in \mathcal{A}$ specifies a scaling factor applied to the bid at time $t$. After taking action $\boldsymbol{a}_t$, the auto-bidding agent obtains a reward $r_t(\boldsymbol{s}_t, \boldsymbol{a}_t) \in \mathcal{R}$ reflecting the value of won impressions during $[t, t+1)$, and incurs a cost $c_t(\boldsymbol{s}_t, \boldsymbol{a}_t)$ that corresponds to the expenditure within this period. The environment dynamics are characterized by $\mathcal{P}(\cdot|\boldsymbol{s}_t, \boldsymbol{a}_t)$ that governs the evolution of the state, and $\gamma \in [0, 1]$ is the discount factor.

The goal of auto-bidding is to find a policy $\pi_{\boldsymbol{\theta}}(\cdot|\boldsymbol{s})$ maximizing the expected cumulative reward while satisfying the budget constraint $B$, formulated as:

$$\mathcal{L}(\boldsymbol{\theta}) = -\mathbb{E}_{\boldsymbol{a}_t \sim \pi_{\boldsymbol{\theta}}(\cdot|\boldsymbol{s}_t), \boldsymbol{s}_{t+1} \sim \mathcal{P}(\cdot|\boldsymbol{s}_t, \boldsymbol{a}_t)} \Big[ \sum_{t=1}^{T} \gamma^t r_t(\boldsymbol{s}_t, \boldsymbol{a}_t) \Big], \quad \text{s.t.} \sum_{t=1}^{T} c_t(\boldsymbol{s}_t, \boldsymbol{a}_t) \leq B. \quad (1)$$

**Generative Auto-bidding.** Recent studies have demonstrated the effectiveness of the generative auto-bidding paradigm over traditional reinforcement learning in improving bidding performance (Guo et al., 2024; Li et al., 2025). This paradigm generates bidding trajectories through conditional generative modeling, enabling flexible and effective policy learning. Let $\mathcal{D}$ denote the offline dataset of trajectories $\tau$ and their quality $y(\tau)$. The generative auto-bidding objective is:

$$\mathcal{L}(\boldsymbol{\theta}) = -\mathbb{E}_{(\tau, y(\tau)) \sim \mathcal{D}}[\log p_{\boldsymbol{\theta}}(\tau|y(\tau))], \quad (2)$$

where $p_{\boldsymbol{\theta}}$ denotes the likelihood of trajectories conditioned on their quality signals. Building on the advanced paradigm, we propose a multi-task auto-bidding generative framework to better address diverse auto-bidding tasks.

**Multi-task Learning.** MTL aims to train a unified model capable of simultaneously fulfilling $K$ different tasks. The ultimate goal of MTL is to achieve superior performance across all tasks. A trivial method is to optimize the average loss across all tasks:

$$\min_{\boldsymbol{\theta} \in \boldsymbol{\Theta}} \frac{1}{K} \sum_{k=1}^{K} \mathcal{L}_k(\boldsymbol{\theta}), \quad (3)$$

where $\mathcal{L}_k(\boldsymbol{\theta})$ is the task-specific loss function associated with the $k$-th task.

However, directly minimizing the average loss often leaves some tasks under-optimized due to scale and difficulty imbalances as well as gradient interference. Loss-based methods instead minimize a weighted sum of task losses $\sum_{k=1}^{K} w_k \mathcal{L}_k(\boldsymbol{\theta})$ with $w_k \geq 0$, where $\boldsymbol{\theta}$ collects shared and task-specific parameters. The weights are typically set based on uncertainty (Kendall et al., 2018), learning pace (Murugesan & Carbonell, 2017; Liu et al., 2019; 2023), random loss weight (Lin et al., 2021), or task prioritization (Guo et al., 2018) to balance optimization across tasks. Gradient-based methods modify task gradients on the shared network using gradient information, e.g., normalization (Chen et al., 2018b), projection (Yu et al., 2020), or conflict mitigation (Liu et al., 2021a). However, both families are driven by training dynamics and tend to overfit transient signals (Mao et al., 2022), which leads to poor generalization under distribution shift and in volatile bidding environments, and to misalignment with validation time objectives.

## 3 METHOD

This section presents a unified multi-task learning framework for auto-bidding, as shown in Fig. 2. We first introduce a validation-aligned multi-task optimization strategy, then propose a temporal module that captures periodic auction dynamics and integrates it with the generative backbone. Finally, we provide the theoretical analysis to show the convergence of the proposed method.

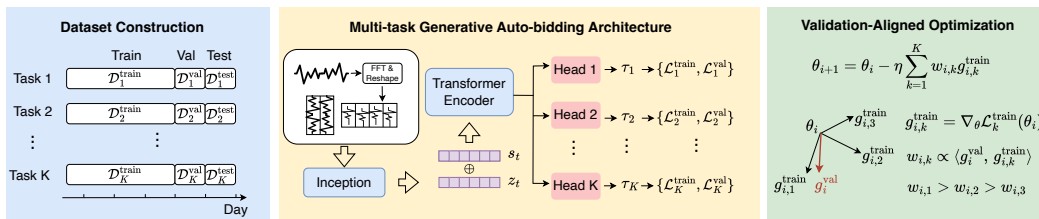

Figure 2: **The overall flowchart of VAMO and multi-task learning architectures.** The online-generated dataset is partitioned into a training and a validation dataset over time, where distribution shift probably happens when the bidding environment changes a lot. The neural architecture extracts the shared information and constitutes task-specific generative auto-bidding modules. VAMO learns to align with the objective of the shifted test environment while balancing multi-task performance.

### 3.1 VALIDATION-ALIGNED MULTI-TASK OPTIMIZATION

Much of prior work in multi-task optimization focuses on mitigating task interference, such as gradient conflicts (Liu et al., 2021a). In our setting, such issues are less pronounced because all bidding tasks ultimately support the same goal of maximizing advertiser value under budget constraints, resulting in coherent optimization signals. Instead, our work addresses a different and practically critical challenge: generalization under temporal distribution shifts in volatile bidding environments. In real-world auto-bidding systems, user behavior evolves quickly and competitor strategies change unpredictably (Borissov et al., 2010), causing frequent and often abrupt distributional shifts that are challenging to forecast (Gao et al., 2025). As a result, strong performance on training data often does not translate to reliable deployment performance.

Furthermore, different tasks exhibit varying sensitivities to these distribution shifts, with some showing short-term effectiveness and others experiencing pronounced deterioration. This disparity underscores the importance of adaptive task weighting that can respond to evolving task-specific generalization trends. Yet, without a reliable estimate of out-of-distribution performance, such adaptation risks overfitting to transient patterns or favoring short-term gains.

**Multi-Task Optimization with Validation Alignment.** To address these challenges, we introduce a validation-aligned optimization strategy that uses a temporally held-out validation set to estimate generalization. In auto-bidding, where data consists of sequential auction logs, we reserve a contiguous time window after the training period as validation. This design avoids temporal leakage, preserves real-world distribution shifts, and provides a reliable signal for adaptive task balancing. The resulting validation loss provides a realistic estimate of future performance and directly informs the adaptive task weighting mechanism during training.

Given this setup, let $\{\mathcal{L}_k^{\text{train}}(\boldsymbol{\theta})\}_{k=1}^K$ and $\{\mathcal{L}_k^{\text{val}}(\boldsymbol{\theta})\}_{k=1}^K$ denote the per-task training and validation losses. Our ultimate goal is to minimize the average validation loss across tasks:

$$\mathcal{L}^{\text{val}}(\boldsymbol{\theta}) = \frac{1}{K}\sum_{k=1}^K \mathcal{L}_k^{\text{val}}(\boldsymbol{\theta}), \tag{4}$$

which directly targets improved generalization in the downstream application. During training, we use an adaptive weighted training loss as a surrogate: $\mathcal{L}^{\text{train}}(\mathbf{w}, \boldsymbol{\theta}) = \sum_{k=1}^K w_k \mathcal{L}_k^{\text{train}}(\boldsymbol{\theta})$, where the weights $\mathbf{w} = \{w_1, w_2, \cdots, w_K\} \in \Delta^{K-1}$ lie in $(K-1)$-dimensional probability simplex, i.e., $w_k \geq 0$ and $\sum_{k=1}^K w_k = 1$. Starting from parameters $\boldsymbol{\theta}_i$ at the $i$-th iteration, a one-step update with step size $\eta > 0$ yields:

$$\boldsymbol{\theta}_{i+1}(\mathbf{w}) = \boldsymbol{\theta}_i - \eta \sum_{k=1}^K w_{i,k}\, \boldsymbol{g}_{i,k}^{\text{train}}, \qquad \boldsymbol{g}_{i,k}^{\text{train}} \triangleq \nabla_{\boldsymbol{\theta}} \mathcal{L}_k^{\text{train}}(\boldsymbol{\theta}_i). \tag{5}$$

The $w_{i,k}$ are adaptively adjusted using validation feedback to maximize improvements in $\mathcal{L}^{\text{val}}(\boldsymbol{\theta})$.

**Analysis of Validation Loss Change.** Using a first-order Taylor expansion of $\mathcal{L}^{\text{val}}(\boldsymbol{\theta})$ around $\boldsymbol{\theta}_i$, the validation loss after one update is approximated as:

$$\mathcal{L}^{\text{val}}(\boldsymbol{\theta}_{i+1}) = \mathcal{L}^{\text{val}}(\boldsymbol{\theta}_i) - \eta \Big\langle \boldsymbol{g}_i^{\text{val}}, \sum_{k=1}^K w_{i,k} \boldsymbol{g}_{i,k}^{\text{train}} \Big\rangle + \mathcal{O}(||\boldsymbol{\theta}_{i+1} - \boldsymbol{\theta}_i||), \tag{6}$$

---

**Algorithm 1:** Validation Aligned Multi-Task Optimization (VAMO)

1: **Input**: Maximum iteration number $I$; Learning rate $\eta$; Temperature hyperparameter $\lambda$; Task number $K$; Batch size $\mathcal{B}$; Dataset $\mathcal{D}$;

2: Let $d_{\max}$ be the last day in $\mathcal{D}$. Split the data into training and validation sets by time: $\mathcal{D}^{\text{train}} = \bigcup_{d < d_{\max}} \mathcal{D}_d$, $\mathcal{D}^{\text{val}} = \mathcal{D}_{d_{\max}}$;

3: Initialize model parameters $\boldsymbol{\theta}_0$;

4: **for** $i = 0 : I$ **do**

5:     Sample a training mini-batch $B^{\text{train}} \subset \mathcal{D}^{\text{train}}$ with $|B^{\text{train}}| = \mathcal{B}$, and a validation mini-batch $B^{\text{val}} \subset \mathcal{D}^{\text{val}}$ with $|B^{\text{val}}| = \mathcal{B}$, with task proportions matched to the empirical distribution over tasks in $\mathcal{D}$.

6:     Compute $\boldsymbol{g}_{i,k}^{\text{train}} = \nabla_{\boldsymbol{\theta}} \mathcal{L}_k^{\text{train}}(\boldsymbol{\theta}_i; B_k^{\text{train}})$ for each task, where $B_k^{\text{train}} \subset B^{\text{train}}$;

7:     Compute $\boldsymbol{g}_i^{\text{val}} = \nabla_{\boldsymbol{\theta}} \mathcal{L}^{\text{val}}(\boldsymbol{\theta}_i; B^{\text{val}})$;

8:     Compute marginal gains $m_{i,k} = \langle \boldsymbol{g}_i^{\text{val}}, \boldsymbol{g}_{i,k}^{\text{train}} \rangle$;

9:     Compute weights $w_{i,k} = \frac{\exp(m_{i,k}/\lambda)}{\sum_{j=1}^K \exp(m_{i,j}/\lambda)}$;

10:     Update parameters $\boldsymbol{\theta}_{i+1} = \boldsymbol{\theta}_i - \eta \sum_k w_{i,k} \boldsymbol{g}_{i,k}^{\text{train}}$;

11: **end for**

---

where $\boldsymbol{g}_i^{\text{val}} \triangleq \nabla_{\boldsymbol{\theta}} \mathcal{L}^{\text{val}}(\boldsymbol{\theta}_i)$. Consequently, the *change* in validation loss is:

$$\Delta\mathcal{L}^{\text{val}} \triangleq \mathcal{L}^{\text{val}}(\boldsymbol{\theta}_{i+1}) - \mathcal{L}^{\text{val}}(\boldsymbol{\theta}_i) \approx -\eta \sum_{k=1}^K w_{i,k} m_{i,k}, \tag{7}$$

where $m_{i,k} \triangleq \langle \boldsymbol{g}_i^{\text{val}}, \boldsymbol{g}_{i,k}^{\text{train}} \rangle$ denotes the *marginal gain* of the overall performance from task $k$.

The marginal gain $m_{i,k}$ provides a first-order measure to quantify task $k$'s contribution to the reduction of the validation loss. Intuitively, a positive $m_{i,k} > 0$ indicates that the training gradient of task $k$ is aligned with the direction of validation improvement, and thus increasing its weight promotes generalization. Conversely, a negative $m_{i,k} < 0$ suggests misalignment or conflict with validation dynamics, and down-weighting such tasks avoids harmful interference during training. This establishes a principled weighting scheme that directly links each task's weight to its marginal gain in validation performance. In particular, learning weights to maximize $\sum_{k=1}^K w_{i,k} m_{i,k}$ over the probability simplex $\Delta^{K-1}$ aligns the training update with the steepest predicted decrease in validation loss, prioritizing the task that contributes most to generalization.

**Balanced Optimization with Entropy-Regularization.** While selecting the task with the highest marginal gain improves validation performance in the short term, this greedy approach often leads to imbalanced optimization, where one task dominates and others are neglected. Such imbalance may compromise training stability and weaken the effectiveness of multi-task learning. To promote balanced task participation, we introduce entropy regularization into the objective:

$$\max_{\mathbf{w} \in \Delta^{K-1}} \sum_{k=1}^K w_{i,k} m_{i,k} + \lambda \mathcal{H}(\mathbf{w}), \qquad \mathcal{H}(\mathbf{w}) = -\sum_{k=1}^K w_{i,k} \log w_{i,k}, \tag{8}$$

where $\lambda > 0$ controls the strength of regularization. This convex optimization has a closed solution:

$$w_{i,k}^* = \frac{\exp(m_{i,k}/\lambda)}{\sum_{j=1}^K \exp(m_{i,j}/\lambda)}. \tag{9}$$

It can be seen that $w_{i,k}^* = \text{softmax}(m_{i,k}/\lambda)$ with $\lambda$ as a temperature hyperparameter. A smaller $\lambda$ produces sharper distributions, with weights concentrating around tasks with the highest marginal gain, while a larger $\lambda$ flattens the distribution, approaching uniform allocation. The entropy term thus balances validation alignment with optimization stability, promoting robust and effective MTL.

## 3.2 MULTI-TASK GENERATIVE AUTO-BIDDING ARCHITECTURE

Besides optimization, the model architecture is also important in multi-task learning. We adopt a shared-bottom architecture augmented with task-specific generators. The shared backbone extracts

common representations across tasks, promoting knowledge transfer and parameter efficiency. Each task-specific generator then maps these shared features to its own output space, enabling flexible and specialized modeling of different bidding objectives.

**Shared Backbone and Task-specific Generators.** The backbone in this work depicts the conditional distribution over bidding trajectories. Each bidding trajectory is represented as a sequence of states. At each step $t$, the state embedding $\mathbf{s}_t \in \mathbb{R}^h$ is enriched with a periodicity-aware representation $\boldsymbol{z}_t \in \mathbb{R}^h$ derived from a temporal module, yielding an augmented state vector:

$$\tilde{\mathbf{s}}_t = \mathbf{s}_t + \boldsymbol{z}_t. \tag{10}$$

The Transformer encoder (Vaswani et al., 2017) then maps the historical sequence $\tilde{\mathbf{s}}_{<t} = \{\tilde{\mathbf{s}}_1, \ldots, \tilde{\mathbf{s}}_{t-1}\}$ into a hidden representation $\mathbf{h}_t = f_{\boldsymbol{\theta}^s}(\tilde{\mathbf{s}}_{<t}) \in \mathbb{R}^h$, where $f_{\boldsymbol{\theta}^s}$ denotes the shared encoder. The task-specific head autoregressively generates the next state for task $k$: $p_{\boldsymbol{\theta}}(\boldsymbol{s}_{k,t} \mid \boldsymbol{s}_{<t}, k, y(\tau_k)) = f_{\boldsymbol{\theta}^k}(\mathbf{h}_t, y(\tau_k))$, where $f_{\boldsymbol{\theta}^k}$ denotes the generator specialized for task $k$ and $\boldsymbol{\theta} = \{\boldsymbol{\theta}^s, \boldsymbol{\theta}^k\}$ represents shared parameter and task-specific parameter, respectively. Thus, the conditional distribution over a trajectory $\tau_k = \{s_{k,1}, \ldots, s_{k,T}\}$ is factorized as:

$$p_{\boldsymbol{\theta}}(\tau_k \mid y(\tau_k)) = \prod_{t=1}^{T} p_{\boldsymbol{\theta}}(\boldsymbol{s}_{k,t} \mid \boldsymbol{s}_{<t}, k, y(\tau_k)). \tag{11}$$

**Periodicity-aware Temporal Module.** We observe that all bidding tasks share a common bidding environment, which induces similarities in traffic patterns and market dynamics. The underlying temporal patterns exhibit strong periodicity, as shown in Fig. 1a. To capture this property, we employ a periodicity-aware time series module based on TimesNet (Wu et al., 2023). Given a multivariate history time series $\boldsymbol{x}_{-H:-1} \in \mathbb{R}^{H \times d}$, we first compute its frequency spectrum via Fast Fourier Transform (FFT):

$$S(f) = \left| \sum_{h=1}^{H} \boldsymbol{x}_h e^{-2\pi i f h / H} \right|, \quad f = 0, 1, \ldots, H - 1, \tag{12}$$

and construct a candidate period set $\mathcal{Q}$ from dominant frequencies. For each $q \in \mathcal{Q}$, the time series is reshaped into a two-dimensional period–phase tensor of shape $q \times \lfloor H/q \rfloor$, separating intra-period temporal structures from inter-period evolutionary trends. A parameter-efficient Inception block (Szegedy et al., 2015) is then applied to jointly capture local and global dependencies.

$$\boldsymbol{z}_t = \text{Aggregate}\big(\text{Inception}(\text{Reshape}_q(\boldsymbol{x}_{-H:-1})) : q \in \mathcal{Q}\big). \tag{13}$$

Finally, $\boldsymbol{z}_t$ is fed into the shared backbone of our multi-task framework, enabling all bidding tasks to benefit from a unified, periodicity-aware representation of auction dynamics.

## 3.3 THEORETICAL ANALYSIS

We provide a theoretical analysis of the proposed VAMO strategy. Under the following assumptions, we establish that VAMO converges to a stationary point and derive sublinear convergence rates, offering theoretical justification for its reliable performance.

**Assumption 1** (**Smoothness**). *The validation loss $\mathcal{L}^{\text{val}}$ is $L$-smooth, i.e., there exists a positive real constant $L$ to satisfy $|\mathcal{L}^{\text{val}}(\boldsymbol{\theta}_i) - \mathcal{L}^{\text{val}}(\boldsymbol{\theta}_j)| \leq L\|\boldsymbol{\theta}_i - \boldsymbol{\theta}_j\|_2 \ \forall \ \boldsymbol{\theta}_i \text{ and } \boldsymbol{\theta}_j$.*

**Assumption 2** (**Bounded gradients**). *There exists $G > 0$ such that for all tasks $k$ and iterations $i$, $\|\boldsymbol{g}_{i,k}^{\text{train}}\|_2 \leq G$.*

**Assumption 3** (**Alignment coverage**). *At each iteration $i$, the convex cone spanned by the $K$ training task gradients provides sufficient coverage of the validation direction. Concretely, there exist constants $\gamma \in (0, 1]$ and $M \geq 1$ such that:*

$$\max_{\mathbf{w} \in \Delta^{K-1}} \ \left\langle \boldsymbol{g}_i^{\text{val}}, \sum_{k=1}^{K} w_{i,k} \boldsymbol{g}_{i,k}^{\text{train}} \right\rangle \geq \gamma \|\boldsymbol{g}_i^{\text{val}}\|_2^2, \qquad \min_{\mathbf{w} \in \Delta^{K-1}} \ \frac{\|\sum_{k=1}^{K} w_{i,k} \boldsymbol{g}_{i,k}^{\text{train}}\|_2}{\|\boldsymbol{g}_i^{\text{val}}\|_2} \ \leq \ M.$$

The assumption indicates that there exists a convex combination of training gradients with nontrivial positive alignment with $\boldsymbol{g}_i^{\text{val}}$ and comparable magnitude. A larger $\gamma$ and a smaller $M$ imply better alignment. The requirement is mild and only excludes cases where the training gradients are nearly orthogonal to the validation gradient or have extreme norm mismatch.

**Lemma 1** (Maximal alignment among convex combinations). *Let* $m_{i,k} \triangleq \langle \boldsymbol{g}_i^{\mathrm{val}}, \boldsymbol{g}_{i,k}^{\mathrm{train}} \rangle$ *and* $m_i = (m_{i,1}, \ldots, m_{i,K})^\top \in \mathbb{R}^K$. *Let* $\boldsymbol{d}_i^\star \in \arg\max_{\mathbf{w} \in \Delta^K} \langle \boldsymbol{g}_i^{\mathrm{val}}, \sum_k w_{i,k} \boldsymbol{g}_{i,k}^{\mathrm{train}} \rangle$, $d_i = \sum_k w_{i,k}^\lambda \boldsymbol{g}_{i,k}^{\mathrm{train}}$ *where* $w_i^\lambda = \mathrm{softmax}(m_i/\lambda)$ *for some* $\lambda > 0$. *Then*

$$\langle \boldsymbol{g}_i^{\mathrm{val}}, d_i \rangle \geq \langle \boldsymbol{g}_i^{\mathrm{val}}, \boldsymbol{d}_i^\star \rangle - \lambda \log K.$$

**Theorem 1** (Convergence). *Under Assumptions 1/2/3 and Lemma 1, and the update is* $\boldsymbol{\theta}_{i+1} = \boldsymbol{\theta}_i - \eta \, d_i$, *for any fixed step size* $\eta > 0$ *and* $I \geq 1$, *we have:*

$$\frac{1}{I} \sum_{i=0}^{I-1} \mathbb{E}\big[\|\boldsymbol{g}_i^{\mathrm{val}}\|_2^2\big] \leq \frac{\mathbb{E}\big[\mathcal{L}^{\mathrm{val}}(\boldsymbol{\theta}_0) - \inf_{\boldsymbol{\theta}} \mathcal{L}^{\mathrm{val}}(\boldsymbol{\theta})\big]}{\eta \, \gamma \, I} + \underbrace{\frac{\lambda \log K}{\gamma}}_{\text{entropy floor}} + \underbrace{\frac{LG^2}{2\gamma} \eta}_{\text{step size floor}}. \qquad (14)$$

*As* $I \to \infty$, *the average squared validation gradient norm converges to a neighborhood of radius* $O(\lambda) + O(\eta)$.

**Corollary 1.** *Under the Robbins-Monro conditions on the step size* $\eta_i$, *i.e.,* $\sum_{i=0}^\infty \eta_i = \infty$ *and* $\sum_{i=0}^\infty \eta_i^2 < \infty$, *and with* $\lambda = 0$ *(hard-max weights) or* $\lambda = \boldsymbol{\theta}(\eta)$, *then* $\lim_{I \to \infty} \frac{1}{I} \sum_{i=0}^{I-1} \mathbb{E}\big[\|\boldsymbol{g}_i^{\mathrm{val}}\|_2^2\big] = 0$. *This establishes convergence to a first-order stationary point in the ergodic sense, with a sublinear rate of* $O(1/I)$.

## 4 EXPERIMENTS

### 4.1 EXPERIMENTAL SETUP

**Experiment Environment.** We evaluate our method on both simulated and real-world scenarios to demonstrate its effectiveness. Our experiments evaluate three bidding tasks, each targeting a specific campaign objective: Store conversion bidding typically aims at increasing ad-driven store-wide Gross Merchandise Value (GMV), direct conversion bidding aims at improving directly ad-driven product GMV (Dir-GMV), and add-to-cart bidding aims at boosting the number of ad-driven add-to-cart actions (CartCnt). The simulated experiments are conducted in an open-source advertising system as used in Guo et al. (2024). The real-world experiments are conducted on one of the world's largest E-commerce platforms, TaoBao. Detailed settings are given in Appendix C.

**Baselines.** We compare our validation-aligned approach VAMO against three categories of baselines: single-task learning, loss-based methods, and gradient-based methods. The single-task learning (STL) trains an independent model for each task separately. The loss-based methods consist of the vanilla approach that assigns equal weights to all tasks, Dynamic Weight Average (DWA) (Liu et al., 2019), which adaptively adjusts task weights based on rates of loss changes, and FAMO (Liu et al., 2023), which balances task losses by ensuring each task's loss decreases approximately at an equal rate. The gradient-based methods include PCGrad (Yu et al., 2020), which projects conflicting gradients to mitigate interference, and FairGrad (Ban & Ji, 2024), which adjusts gradients through fair resource allocation to ensure balanced task updates.

**Implementation Details.** We used a 10-day dataset, with the first 8 days allocated to training data, day 9 reserved for validation data to adjust loss weights, and the last day serving as the test set to evaluate model performance. The temperature hyperparameter is set to 1.

**Evaluations.** The primary evaluation metrics for three bidding tasks are GMV, Direct GMV (Dir-GMV), and Add-to-Cart Count (CartCnt). In addition, we adopt a common metric for evaluating multi-task learning (MTL) performance: $\Delta m\%$, which measures the average per-task performance drop relative to single-task learning (STL) (Liu et al., 2023; Shen et al., 2024). It is calculated by $\Delta m\% = \frac{1}{K} \sum_{k=1}^K -(M_{m,k} - M_{b,k})/M_{b,k} \times 100$, where $M_{b,k}$ and $M_{m,k}$ are the STL and $m$'s value for metric $M_k$. A more negative $\Delta m\%$ indicates stronger MTL performance. In online experiments, we extend the evaluations by incorporating three efficiency metrics: ROI (=GMV/COST), Dir-ROI (=Dir-GMV/COST), and COST-per-Cart (=COST/CartCnt). These supplementary metrics provide a more comprehensive view of campaign effectiveness, where higher ROI/Dir-ROI and lower COST-per-Cart are preferred.

Table 1: **Results on three bidding tasks in the simulation environment.** Each experiment is conducted across three random seeds, and the mean is reported. Metrics include return for each task and overall MTL performance $\Delta m\%$. The best result is marked in bold. The $\downarrow$ denotes the lower the better.

| Method | Store Conversion | Direct Conversion | Add-to-Cart | $\Delta m\%\downarrow$ |
|---|---|---|---|---|
| STL | 12.06 | 17.88 | 2.87 | - |
| Vanilla | 17.67 | 23.72 | 2.92 | -26.97 |
| DWA (Liu et al., 2019) | 18.42 | 20.64 | 2.55 | -19.01 |
| FAMO (Liu et al., 2023) | 18.68 | 19.56 | 3.31 | -26.54 |
| PCGrad (Yu et al., 2020) | 17.12 | 21.92 | 1.94 | -10.72 |
| FairGrad (Ban & Ji, 2024) | 15.18 | 19.49 | 2.40 | -6.17 |
| **VAMO (Ours)** | **24.23** | **24.25** | **3.77** | **-55.97** |

Table 2: **Results on three bidding tasks in real-world A/B tests.** We compare against the Vanilla baseline only. Thus, $\Delta m\%$ is not reported.

| Method | Store Conversion | | Direct Conversion | | Add-to-Cart | |
|---|---|---|---|---|---|---|
| | GMV $\uparrow$ | ROI $\uparrow$ | Dir-GMV $\uparrow$ | Dir-ROI $\uparrow$ | CartCnt $\uparrow$ | COST-per-CartCnt $\downarrow$ |
| Vanilla | 200 | 2.50 | 235 | 2.53 | 8.62 | 6.76 |
| **VAMO (Ours)** | **205** | **2.58** | **246** | **2.67** | **8.84** | **6.64** |
| Diff | +2.5% | +3.3% | + 4.6% | +5.4% | +2.5% | -1.8% |

## 4.2 EMPIRICAL RESULT ANALYSIS

We provide performance comparisons on simulated experiments in Table 1. Our method achieves the best overall MTL performance among both gradient-based and loss-based methods, and also delivers the best results on each individual task. We observe that most MTL baselines outperform single-task learning, which contrasts with common observations in the literature that MTL may suffer from performance degradation due to task interference (Yu et al., 2020; Chen et al., 2020; Liu et al., 2021a; 2023; Ban & Ji, 2024). This discrepancy may be attributed to the fact that all tasks in our setting belong to the auto-bidding domain, with similar input spaces and temporal dynamics, which could reduce task conflict. Additionally, since single-task models are trained on limited data, joint training may improve generalization by enabling more efficient data utilization and knowledge sharing across related tasks. The strong performance of our method stems from its adaptive weighting mechanism, which leverages validation alignment to guide task gradient updates. Unlike baselines that solely rely on training dynamics, our approach aligns task weights with a held-out validation signal, promoting generalization. The combination of this alignment with entropy regularization avoids focusing on one task, resulting in more reliable multi-task learning.

We also conduct online experiments to evaluate the effectiveness of our method in a real-world auto-bidding system, as shown in Table 2. Due to the high operational costs and potential business risks associated with online experiments, we restrict the comparison to the Vanilla baseline and our proposed approach. To protect the privacy of advertisers, all absolute values in the online experiments are uniformly normalized. While the normalized absolute values do not reflect actual magnitudes, the relative improvements (e.g., percentage gains) remain statistically accurate and meaningful for comparison. The online results show that our method achieves 2.5%, 4.6%, and 2.5% improvements in GMV, Dir-GMV, and CartCnt, respectively, confirming the real-world effectiveness of our method.

## 4.3 ABLATION STUDY

**Effects of Validation Signal.** To evaluate the role of the validation set in our task weighting strategy, we conduct an ablation study by removing the held-out validation data and instead using the total training gradient as the alignment target. This variant, denoted as "w/o held-out validation", relies solely on training dynamics without external generalization feedback. As shown in Fig. 3, despite better performance on the store conversion task, it underperforms our validation-aligned method by

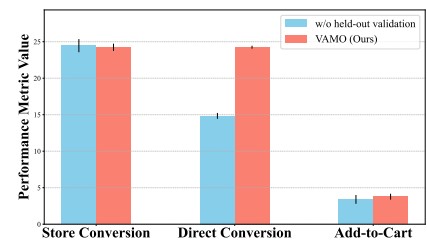

Figure 3: **Ablation on validation signal.** Error bars denote the standard deviation (3 runs).

21.21 in overall performance $\Delta m\%$. This discrepancy arises because tasks with rapid loss reduction may dominate the training gradients, biasing optimization toward certain tasks. While beneficial in the short term, this leads to overfitting to transient patterns and harms overall generalization. In contrast, our method uses out-of-distribution feedback from a temporally separated hold-out set, providing a more reliable and realistic estimate of a task's generalization impact. This enables robust and well-calibrated task balance, leading to improved generalization.

**Influence of $\lambda$.** To investigate the role of entropy regularization, we conduct an ablation study by varying the temperature parameter $\lambda$, which governs the strength of regularization. We set $\lambda$ from near-zero to very large, corresponding to different levels of entropy regularization. As $\lambda \to 0$, the weighting becomes assigning all mass to the task with the highest marginal gain. As $\lambda \to \infty$, the weights converge to uniform, which corresponds to the *Vanilla* baseline in our experiments. As shown in Fig. 4, $\lambda = 0.1$ reduces to greedy task weighting, where only the task with the highest marginal gain dominates training. This leads to unstable optimization and poor generalization due to task imbalance. When $\lambda \to \infty$, the vanilla

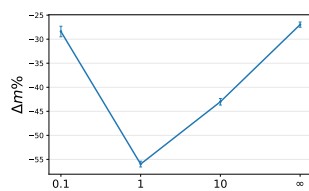

Figure 4: **Ablation on $\lambda$.** The error bars denote the standard deviation (3 runs).

baseline treats all tasks equally regardless of their impact on validation performance. This approach fails to prioritize high-impact tasks, limiting its ability to adapt to dynamic bidding environments. Moderate values can achieve optimal performance, striking a trade-off between validation alignment and task balance. Our method VAMO allows the model to dynamically emphasize tasks that contribute most to validation improvement, while maintaining sufficient balance in training updates to ensure robustness.

**Effect of Periodicity-aware Temporal Module.** To evaluate the effectiveness of our periodicity-aware temporal module, we conduct an ablation study with two variants: (i) *without TimesNet*, where the module is removed; (ii) *with LSTM*, where an LSTM model is used to model temporal dependencies, serving as a baseline for sequential modeling. Both variants maintain the same multi-task

Table 3: **Ablation on temporal modeling.** Performance comparison of no temporal module, LSTM variant, and our periodicity-aware design.

| Architecture | Store Conversion | Direct Conversion | Add-to-Cart | $\Delta m\% \downarrow$ |
|---|---|---|---|---|
| w/o TimesNet | 18.01 | 16.20 | 3.70 | -22.95 |
| with LSTM | **25.15** | 17.36 | 3.75 | -45.43 |
| **VAMO (Ours)** | 24.23 | **24.25** | **3.77** | **-55.97** |

architecture and training pipeline to ensure a fair comparison. Results are reported in Table 3. Removing the temporal module leads to a performance drop, highlighting the critical role of temporal modeling. While LSTM outperforms our method on a specific task, its overall performance lags behind, demonstrating that although LSTM can capture general sequential patterns, it falls short in modeling the multi-periodic structures inherent in auction dynamics. Additionally, the LSTM variant shows improved performance over the baselines listed in Table 1, further highlighting the effectiveness and necessity of our approach.

## 5 CONCLUSION

**Technical Discussion.** This work focuses on the distributional shift in multi-task learning for online auto-bidding. Our proposed VAMO addresses the issue by adaptively balancing tasks based on validation signals, improving generalization under nonstationary environments. To capture shared temporal dynamics, we incorporate a dedicated module modeling multi-scale periodicity in the environment. The resulting multi-task framework enables robust and transferable learning across multiple auto-bidding tasks. The theoretical analysis establishes convergence guarantees for the proposed VAMO scheme, offering insights into its stable dynamics under nonstationary distributions. The significant improvement in multi-task performance over both single-task models and baselines validates the effectiveness of our method and practical value in real-world auto-bidding systems.

**Limitations and Future Work.** This work has validated the effectiveness of the proposed approach on multi-task generative auto-bidding. Future work will explore enhancing the multi-task model with more complex architectures, such as mixture-of-experts models.

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

## A  LITERATURE REVIEW

**Auto-bidding Methods.** The mainstream auto-bidding methods can be broadly categorized into two branches to achieve diverse bidding tasks: Reinforcement Learning (RL)-based auto-bidding methods and generative auto-bidding methods. *RL-based auto-bidding methods* model auto-bidding as a Markov Decision Process and learn the optimal bidding policy through RL techniques. For example, Deep Reinforcement Learning to Bid (DRLB) (Wu et al., 2018) uses deep Q-network (DQN) (Mnih et al., 2015) with reward shaping to maximize impression value under budget constraints. USCB (He et al., 2021) employs the DDPG (Silver et al., 2014) algorithm to dynamically adjust bidding parameters to an optimal strategy. SORL (Mou et al., 2022) develops a variance-suppressed conservative Q-learning method to effectively learn auto-bidding policies. Due to the risks of real-time bidding, offline RL methods such as BCQ (Fujimoto et al., 2019), CQL (Kumar et al., 2020), and IQL (Kostrikov et al., 2022) have gained prominence for learning policies solely from historical datasets without online interaction.

*Generative auto-bidding methods* show greater potential than RL-based methods because they can better model the complex distribution of bidding strategies. These methods formulate auto-bidding as a conditional generative modeling problem. Decision Transformer (DT) (Chen et al., 2021) generates optimal actions using an auto-regressive transformer conditioned on desired returns, historical states, and action. GAS (Li et al., 2025) adopts DT to generate actions for auto-bidding and employs a Monte Carlo Tree Search-inspired post-training refinement to better align generated bids with diverse user preferences. AIGB (Guo et al., 2024) employs a conditional diffusion model to generate bidding trajectories alongside an inverse dynamic model for action generation. In this work, we focus on generative auto-bidding methods and aim to develop a unified framework to handle multiple bidding tasks simultaneously.

**Multi-task Learning.** Multi-task learning aims to jointly learn multiple tasks within a single model, improving learning efficiency by enabling information sharing across tasks (Caruana, 1997; Sun et al., 2020; Xu et al., 2020; Thung & Wee, 2018; Yang et al., 2022). Common architectural approaches include the shared bottom model, which employs a shared backbone with separate task-specific heads. More advanced variants include cross-stitch networks (Misra et al., 2016), which learn adaptive feature sharing between tasks, and the Multi-Task Attention Network (MTAN) (Liu et al., 2019), which uses soft attention to dynamically select shared features for each task. Other representative architectures are MMoE (Ma et al., 2018) and PLE (Tang et al., 2020), designed to balance shared and task-specific representations effectively.

For multi-task optimization, approaches can be roughly divided into gradient-based and loss-based methods. *Gradient-based methods* balance tasks by manipulating gradients, including Pareto optimal solutions (Désidéri, 2012; Sener & Koltun, 2018), gradient normalization (Chen et al., 2018b), gradient projection (Yu et al., 2020; Liu et al., 2021a;b), gradient sign dropout (Chen et al., 2020), and Nash bargaining solution (Navon et al., 2022). *Loss-based methods* adaptively adjust task-specific loss weights during training to balance learning progress among tasks. Representative approaches include uncertainty weighting (Kendall et al., 2018), random loss weighting (Lin et al., 2021), and strategies based on learning dynamics (Liu et al., 2019; 2023; Shen et al., 2024). However, these methods rely solely on training signals, which may not reflect true generalization performance and thus limit their ability to generalize under nonstationary environments.

## B  THEOREMS & PROOFS

### B.1  ASSUMPTIONS

**Assumption 1 (Smoothness).** *The validation loss $\mathcal{L}^{\text{val}}$ is L-smooth, i.e., there exists a positive real constant $L$ to satisfy $|\mathcal{L}^{\text{val}}(\boldsymbol{\theta}_i) - \mathcal{L}^{\text{val}}(\boldsymbol{\theta}_j)| \leq L||\boldsymbol{\theta}_i - \boldsymbol{\theta}_j||_2 \ \forall \ \boldsymbol{\theta}_i \ and \ \boldsymbol{\theta}_j.$*

**Assumption 2 (Bounded gradients).** *There exists $G > 0$ such that for all tasks $k$ and iterations $i$, $||\boldsymbol{g}_{i,k}^{\text{train}}||_2 \leq G.$*

**Assumption 3 (Alignment coverage).** *At each iteration $i$, the convex cone spanned by the $K$ training task gradients provides sufficient coverage of the validation direction. Concretely, there exist*

*constants* $\gamma \in (0,1]$ *and* $M \geq 1$ *such that:*

$$\max_{\mathbf{w} \in \Delta^{K-1}} \left\langle \boldsymbol{g}_i^{\text{val}}, \sum_{k=1}^{K} w_{i,k} \, \boldsymbol{g}_{i,k}^{\text{train}} \right\rangle \geq \gamma \, \|\boldsymbol{g}_i^{\text{val}}\|_2^2, \qquad \min_{\mathbf{w} \in \Delta^{K-1}} \frac{\left\| \sum_{k=1}^{K} w_{i,k} \, \boldsymbol{g}_{i,k}^{\text{train}} \right\|_2}{\|\boldsymbol{g}_i^{\text{val}}\|_2} \; \leq \; M.$$

### B.2    PROOF OF LEMMA 1

**Lemma 1** (Maximal alignment among convex combinations) *Let* $m_{i,k} \triangleq \langle \boldsymbol{g}_i^{\text{val}}, \boldsymbol{g}_{i,k}^{\text{train}} \rangle$ *and* $m_i = (m_{i,1}, \ldots, m_{i,K})^\top \in \mathbb{R}^K$. *Let* $\boldsymbol{d}_i^\star \in \arg\max_{\mathbf{w} \in \Delta^K} \langle \boldsymbol{g}_i^{\text{val}}, \sum_k w_{i,k} \boldsymbol{g}_{i,k}^{\text{train}} \rangle$, $d_i = \sum_k w_{i,k}^\lambda \boldsymbol{g}_{i,k}^{\text{train}}$ *where* $w_i^\lambda = \text{softmax}(m_i/\lambda)$ *for some* $\lambda > 0$. *Then*

$$\langle \boldsymbol{g}_i^{\text{val}}, \boldsymbol{d}_i \rangle \; \geq \; \langle \boldsymbol{g}_i^{\text{val}}, \boldsymbol{d}_i^\star \rangle \; - \; \lambda \log K.$$

*Proof.* Consider the entropy-regularized problem

$$(\text{P}_\lambda) \qquad \max_{\mathbf{w} \in \Delta^K} \; \Phi_\lambda(\mathbf{w}; m_i) \triangleq \sum_{k=1}^{K} w_{i,k} \, m_{i,k} \; + \; \lambda \, H(\mathbf{w}), \qquad H(\mathbf{w}) = -\sum_{k=1}^{K} w_{i,k} \log w_{i,k},$$

and the unregularized problem

$$(\text{P}_0) \qquad \max_{\mathbf{w} \in \Delta^K} \; \sum_{k=1}^{K} w_{i,k} \, m_{i,k} \; = \; \max_k m_{i,k}.$$

We proceed in three steps.

Step 1: Entropy-regularized maximization and its value. $d_i = \sum_k w_{i,k}^\lambda g_{i,k}^{\text{train}}$ is the entropy-regularized solution with weights $w_{i,k}^\lambda = \exp(m_{i,k}/\lambda) / \sum_j \exp(m_{i,j}/\lambda)$. Plugging $\boldsymbol{w}^\lambda$ back into $\Phi_\lambda$ yields the optimal value

$$\max_{\mathbf{w} \in \Delta^K} \Phi_\lambda(\mathbf{w}; m_i) \; = \; \lambda \log \sum_{j=1}^{K} \exp\left(m_{i,j}/\lambda\right). \tag{15}$$

Step 2: Log-sum-exp sandwich. For any vector $x \in \mathbb{R}^K$ and $\lambda > 0$,

$$\max_j x_j \; \leq \; \lambda \log \sum_{j=1}^{K} e^{x_j/\lambda} \; \leq \; \max_j x_j \; + \; \lambda \log K. \tag{16}$$

The left inequality follows since $\sum_j e^{x_j/\lambda} \geq e^{\max_j x_j/\lambda}$. For the right inequality, note $\sum_j e^{x_j/\lambda} \leq K \, e^{\max_j x_j/\lambda}$. The *left inequality* is what we use in the proof. The *right inequality* is not required here but provides intuition: log-sum-exp is always within $\lambda \log K$ of the max, meaning entropy regularization yields a smooth approximation of the hard maximum.

Applying equation 16 to $x = m_i$ and using equation 15, we get

$$\max_{\mathbf{w} \in \Delta^K} \sum_k w_{i,k} m_{i,k} \; \leq \; \max_{\mathbf{w} \in \Delta^K} \Phi_\lambda(\mathbf{w}; m_i) \; \leq \; \max_{\mathbf{w} \in \Delta^K} \sum_k w_{i,k} m_{i,k} \; + \; \lambda \log K. \tag{17}$$

Step 3: From optimal value to the inner product at $w_i^\lambda$. At the optimizer $w_i^\lambda$ of $(P_\lambda)$,

$$\sum_{k=1}^{K} w_{i,k}^\lambda m_{i,k} \; = \; \max_{\mathbf{w} \in \Delta^K} \Phi_\lambda(\mathbf{w}; m_i) \; - \; \lambda H(w_i^\lambda). \tag{18}$$

Since $H(w_i^\lambda) \leq \log K$ (Brémaud, 2012), we have

$$\sum_{k=1}^{K} w_{i,k}^\lambda m_{i,k} \; \geq \; \max_{\mathbf{w} \in \Delta^K} \Phi_\lambda(\mathbf{w}; m_i) \; - \; \lambda \log K. \tag{19}$$

Combining equation 19 with the left inequality of equation 17 yields

$$\sum_{k=1}^{K} w_{i,k}^{\lambda} m_{i,k} \;\geq\; \max_{\mathbf{w} \in \Delta^K} \sum_{k} w_{i,k} m_{i,k} \;-\; \lambda \log K. \tag{20}$$

Recalling $m_{i,k} = \langle \boldsymbol{g}_i^{\mathrm{val}}, \boldsymbol{g}_{i,k}^{\mathrm{train}} \rangle$,

$$\langle \boldsymbol{g}_i^{\mathrm{val}}, \boldsymbol{d}_i \rangle = \sum_k w_{i,k}^{\lambda} \langle \boldsymbol{g}_i^{\mathrm{val}}, \boldsymbol{g}_{i,k}^{\mathrm{train}} \rangle \;\geq\; \max_{\mathbf{w} \in \Delta^K} \Big\langle \boldsymbol{g}_i^{\mathrm{val}}, \sum_k w_{i,k} \boldsymbol{g}_{i,k}^{\mathrm{train}} \Big\rangle - \lambda \log K \;=\; \langle \boldsymbol{g}_i^{\mathrm{val}}, \boldsymbol{d}_i^{\star} \rangle - \lambda \log K, \tag{21}$$

which proves the lemma. □

## B.3 Proof of Theorem 1

**Theorem 1** (Convergence) *Under Assumptions 1/2/3 and Lemma 1, and the update is $\boldsymbol{\theta}_{i+1} = \boldsymbol{\theta}_i - \eta\, \boldsymbol{d}_i$, for any fixed step size $\eta > 0$ and $I \geq 1$, we have:*

$$\frac{1}{I} \sum_{i=0}^{I-1} \mathbb{E}\big[\|\boldsymbol{g}_i^{\mathrm{val}}\|_2^2\big] \;\leq\; \frac{\mathbb{E}\big[\mathcal{L}^{\mathrm{val}}(\boldsymbol{\theta}_0) - \inf_{\boldsymbol{\theta}} \mathcal{L}^{\mathrm{val}}(\boldsymbol{\theta})\big]}{\eta\, \gamma\, I} \;+\; \underbrace{\frac{\lambda \log K}{\gamma}}_{\text{entropy floor}} \;+\; \underbrace{\frac{LG^2}{2\gamma}\eta}_{\text{step size floor}}. \tag{22}$$

*As $I \to \infty$, the average squared validation gradient norm converges to a neighborhood of radius $O(\lambda) + O(\eta)$.*

*Proof.* By Assumption 3 and Lemma 1, we have

$$\langle \boldsymbol{g}_i^{\mathrm{val}}, \boldsymbol{d}_i \rangle \geq \max_{\mathbf{w} \in \Delta^K} \Big\langle \boldsymbol{g}_i^{\mathrm{val}}, \sum_k w_{i,k} \boldsymbol{g}_{i,k}^{\mathrm{train}} \Big\rangle - \lambda \log K \geq \gamma \|\boldsymbol{g}_i^{\mathrm{val}}\|_2^2 - \lambda \log K. \tag{23}$$

Apply the second-order Taylor expansion of $\mathcal{L}^{\mathrm{val}}$ around $\boldsymbol{\theta}_i$:

$$\mathcal{L}^{\mathrm{val}}(\boldsymbol{\theta}_{i+1}) = \mathcal{L}^{\mathrm{val}}(\boldsymbol{\theta}_i) - \eta \langle \boldsymbol{g}_i^{\mathrm{val}}, \boldsymbol{d}_i \rangle + \frac{\eta^2}{2} \Big( \sum_{k=1}^{K} w_k \boldsymbol{g}_k^{\mathrm{train}} \Big)^{\top} H^{\mathrm{val}}(\tilde{\boldsymbol{\theta}}) \Big( \sum_{j=1}^{K} w_j \boldsymbol{g}_j^{\mathrm{train}} \Big), \tag{24}$$

where $H^{\mathrm{val}}(\tilde{\boldsymbol{\theta}}) = \nabla_{\boldsymbol{\theta}}^2 \mathcal{L}^{\mathrm{val}}(\tilde{\boldsymbol{\theta}})$ is the Hessian at some point on the line segment between $\boldsymbol{\theta}_i$ and $\boldsymbol{\theta}_{i+1}$. Under Assumptions 1/2, the validation loss is $L$-smooth, i.e. for all $\xi$, $\|H^{\mathrm{val}}(\xi)\|_2 \leq L$, then the magnitude of the second-order remainder is bounded by:

$$\left| \frac{\eta^2}{2} \Big( \sum_k w_k \boldsymbol{g}_k^{\mathrm{train}} \Big)^{\top} H^{\mathrm{val}}(\tilde{\boldsymbol{\theta}}) \Big( \sum_j w_j \boldsymbol{g}_j^{\mathrm{train}} \Big) \right| \leq \frac{L\eta^2}{2} \Big\| \sum_{k=1}^{K} w_k \boldsymbol{g}_k^{\mathrm{train}} \Big\|_2^2. \tag{25}$$

Using the triangle inequality and convexity of the norm,

$$\Big\| \sum_{k=1}^{K} w_k \boldsymbol{g}_k^{\mathrm{train}} \Big\|_2 \leq \sum_{k=1}^{K} w_k \|\boldsymbol{g}_k^{\mathrm{train}}\|_2 \leq G \quad (\text{since } \mathbf{w} \in \Delta^K), \tag{26}$$

so a bound on the second-order remainder is $\frac{L}{2}\eta^2 G^2$. Then,

$$\mathcal{L}^{\mathrm{val}}(\boldsymbol{\theta}_{i+1}) \leq \mathcal{L}^{\mathrm{val}}(\boldsymbol{\theta}_i) - \eta \langle \boldsymbol{g}_i^{\mathrm{val}}, \boldsymbol{d}_i \rangle + \frac{L}{2}\eta^2 G^2 \tag{27a}$$

$$\leq \mathcal{L}^{\mathrm{val}}(\boldsymbol{\theta}_i) - \eta\big( \gamma \|\boldsymbol{g}_i^{\mathrm{val}}\|_2^2 - \lambda \log K \big) + \frac{L}{2}\eta^2 G^2. \tag{27b}$$

Taking the expectation on both sides:

$$\mathbb{E}\big[\mathcal{L}^{\mathrm{val}}(\boldsymbol{\theta}_{i+1})\big] \leq \mathbb{E}\big[\mathcal{L}^{\mathrm{val}}(\boldsymbol{\theta}_i)\big] - \eta\, \gamma\, \mathbb{E}\big[\|\boldsymbol{g}_i^{\mathrm{val}}\|_2^2\big] \;+\; \eta\, \lambda \log K + \frac{L}{2}\eta^2 G^2. \tag{28}$$

Rearranging,

$$\mathbb{E}\big[\|\boldsymbol{g}_i^{\mathrm{val}}\|_2^2\big] \leq \frac{\mathbb{E}\big[\mathcal{L}^{\mathrm{val}}(\boldsymbol{\theta}_i)\big] - \mathbb{E}\big[\mathcal{L}^{\mathrm{val}}(\boldsymbol{\theta}_{i+1})\big]}{\eta\, \gamma} \;+\; \frac{\lambda \log K}{\gamma} + \frac{LG^2}{2\gamma}\eta, \tag{29}$$

summing from $i = 0$ to $I - 1$, the telescoping sum of the first term yields

$$\sum_{i=0}^{I-1} \frac{\mathbb{E}[\mathcal{L}^{\mathrm{val}}(\boldsymbol{\theta}_i)] - \mathbb{E}[\mathcal{L}^{\mathrm{val}}(\boldsymbol{\theta}_{i+1})]}{\eta\,\gamma} = \frac{\mathbb{E}[\mathcal{L}^{\mathrm{val}}(\boldsymbol{\theta}_0)] - \mathbb{E}[\mathcal{L}^{\mathrm{val}}(\boldsymbol{\theta}_I)]}{\eta\,\gamma} \leq \frac{\mathbb{E}[\mathcal{L}^{\mathrm{val}}(\boldsymbol{\theta}_0)] - \inf_{\boldsymbol{\theta}} \mathcal{L}^{\mathrm{val}}(\boldsymbol{\theta})}{\eta\,\gamma}. \tag{30}$$

Therefore,

$$\frac{1}{I}\sum_{i=0}^{I-1} \mathbb{E}\big[\|\boldsymbol{g}_i^{\mathrm{val}}\|_2^2\big] \;\leq\; \frac{\mathbb{E}\big[\mathcal{L}^{\mathrm{val}}(\boldsymbol{\theta}_0) - \inf_{\boldsymbol{\theta}} \mathcal{L}^{\mathrm{val}}(\boldsymbol{\theta})\big]}{\eta\,\gamma\,I} \;+\; \underbrace{\frac{\lambda \log K}{\gamma}}_{\text{entropy floor}} \;+\; \underbrace{\frac{LG^2}{2\gamma}\eta}_{\text{step size floor}}\;, \tag{31}$$

which completes the proof. $\qquad\square$

## C  ADDITIONAL EXPERIMENTAL SETTINGS

We include the number of samples for different tasks under various experimental environments in Table 4. Specifically, we consider the bidding process in a day, where the bidding episode is divided into 96 time steps. Thus, the duration between two adjacent time steps $t$ and $t + 1$ is 15 minutes.

Table 4: Number of samples for different tasks under various experimental environments.

|  | Store Conversion | Direct Conversion | Add-to-Cart |
|---|---|---|---|
| Simulation | 20 | 20 | 10 |
| Real-world | 4700 | 4200 | 1200 |

**Hardware Resource.** The simulated experiments are conducted based on an NVIDIA T4 Tensor Core GPU. We use 10 CPUs and 200G memory.

## D  LLM USAGE

We declare that we use Large Language Models (LLMs) for grammar checking and lexical refinement during the writing process. No LLM-generated content, data analysis, or substantive contributions to the research methodology, results, or conclusions are involved in this work.

