# OpenReview forum: "A Unified Multi-Task Learning Framework for Generative Auto-Bidding with Validation-Aligned Optimization"
_ICLR.cc/2026/Conference — ICLR 2026 Conference Withdrawn Submission_

### Official Review · Reviewer_JoqC · 2025-10-29

**Soundness:** 3
**Presentation:** 4
**Contribution:** 3
**Rating:** 4
**Confidence:** 2

**Summary:**

The paper addresses the problem of auto-bidding in online auctions using a multi-task learning (MTL) framework, where different tasks represent distinct bidding strategies. The authors argue that prevailing MTL methods in this domain, which weight tasks based on training dynamics, generalize poorly to volatile, real-world environments.
To overcome this limitation, the paper introduces VAMO, a novel approach that adaptively re-weights tasks by using gradients from a held-out validation set rather than relying solely on training dynamics. This core contribution, coupled with two modules specifically designed for the auto-bidding problem, is shown to improve bidding performance. The authors support their claims with both theoretical analysis and empirical results.

**Strengths:**

**Clarity and Motivation:** The paper is very well-written, and the authors provide clear, step-by-step motivation for each refinement and design choice that constitutes the final VAMO method.

**Theoretical Grounding:** The proposed method is supported by a comprehensive mathematical framework. The authors provide theoretical results demonstrating guarantees for both convergence and the alignment of validation gradients, which strengthens the paper's methodological contribution.

**Empirical Support:** The empirical results presented in the experiments align well with and help to solidify the paper's theoretical claims.

**Weaknesses:**

**Comparative Theoretical Analysis:** While the theoretical convergence guarantees for VAMO are provided, the paper lacks a comparative analysis of how these guarantees stack up against existing methods. Furthermore, the discussion of convergence is asymptotic, which is theoretically sound but offers limited insight for practitioners. A discussion of the practical convergence rate would be highly beneficial.

**Limited Baselines:** The experimental comparison appears to be limited in scope. Given the extensive body of research on auto-bidding, the small number of baselines makes it difficult to comprehensively assess VAMO's relative performance and situate its contribution within the broader field.

**Statistical Rigor of Experiments:** The empirical evaluation seems to lack statistical robustness. Many results are reported over only three replications, which is very low for drawing strong conclusions. The tables report only mean (w/o standard deviation), which makes it impossible to assess the statistical significance of the claimed improvements over baselines. A more thorough evaluation with an increased number of runs and appropriate statistical significance testing is necessary.

**Parameter Sensitivity:** The ablation study for the $\lambda$ parameter is only conducted over a few values which are widely spaced out. This makes the justification for the final selected value less convincing and suggests the model may be more sensitive to this hyperparameter.

**Limited Datasets:** The experiments appear to be conducted on only one primary dataset. Critically, the baseline methods are not evaluated on the real-world dataset, which is a significant omission. Evaluating all methods on this dataset is crucial for a fair and practical comparison of performance.

**Questions:**

**Performance Discrepancy:** The experimental results indicate that VAMO's performance gains are most pronounced on the "store conversion"  and "direct conversion" metric, while the improvements for "add-to-cart" are more modest. Could you provide some intuition as to why your method might be more impactful for this specific task? Moreover, all these results remain questionable without standard deviations to assess significance.

**Practical Deployment Strategy:** Regarding real-world deployment: the experiments use a fixed 8-day training and 1-day validation split over a 10-week period. In a live setting, the market is highly non-stationary, and models would likely require constant retraining or online adjustment. How do you envision VAMO being deployed in such a dynamic environment? Specifically, how should practitioners demarcate and update the training and validation data over time, and how sensitive is the method to this data-splitting strategy?

---

### Official Review · Reviewer_Jfad · 2025-10-30

**Soundness:** 3
**Presentation:** 2
**Contribution:** 2
**Rating:** 2
**Confidence:** 3

**Summary:**

The paper proposes a way to incorporate continual learning into a generative multi-task auto-bidding model by re-using validation gradients as hints, while incorporating gradient alignment with multi-task learning. The paper also proposes using a module inspired by time-series models of incorporating history into the generative bidding process.

While the basic idea appears to be interesting, this paper requires a significant amount of work before it's ready for publication. See strengths and weaknesses below.

**Strengths:**

1. The problems are important and clearly presented
2. The combination of the ideas together into one system for generative auto-bidding appears new.
3. Benchmarks show improvements.

**Weaknesses:**

1. The idea resembles a vast variety of works in continual learning, and online and stochastic learning with gradient hints. This also includes gradient hints in multi-task learning. See, for example, [1] and derivative works in the continual learning literature, and [2] in the online learning literature that uses the alignment as a gate, and [3] that uses the inner product and the previous gradient as the hint. The work [4] also uses the alignment term, but as a gate instead of a weight. The main novelty of this work is using a different hint, but the idea of optimization with hints is not new.
2. Benchmarking auto-bidding requires an environment that simulates actual ad auctions (with competitors!). It's not clear from the paper how such an experiment is conducted. The experimental setup is very vaguely explained.
3. The paper lacks focus. Two ideas, one of alignment with validation loss, and the other of incorporating seasonality, are merged into one paper. It appears the paper is proposing a system, rather than a machine learning technique, which in my opinion is not well suited to this conference.


---
**References**
[1]: Lopez-Paz, D. and Ranzato, M.A., 2017. Gradient episodic memory for continual learning. Advances in neural information processing systems, 30.
[2]: Dekel, O., Haghtalab, N. and Jaillet, P., 2017. Online learning with a hint. Advances in Neural Information Processing Systems, 30.
[3]: Baydin, A.G., Cornish, R., Rubio, D.M., Schmidt, M. and Wood, F., 2017. Online learning rate adaptation with hypergradient descent. arXiv preprint arXiv:1703.04782.
[4]: Bhaskara, A., Cutkosky, A., Kumar, R. and Purohit, M., 2020, November. Online learning with imperfect hints. In International Conference on Machine Learning (pp. 822-831). PMLR.

**Questions:**

NA

---

### Official Review · Reviewer_mBam · 2025-11-01

**Soundness:** 3
**Presentation:** 3
**Contribution:** 2
**Rating:** 6
**Confidence:** 3

**Summary:**

This paper proposes Validation-Aligned Multi-task Optimization (VAMO), a unified framework for multi-task generative auto-bidding in online advertising. The method addresses the challenge of  heterogeneous advertiser objectives and distribution shifts in volatile bidding environments. Instead of relying purely on training dynamics, VAMO adaptively assigns task weights according to the alignment between per-task training gradients and validation gradients. The model further incorporates a periodicity-aware temporal module based on TimesNet to capture multi-scale temporal regularities and enhance cross-task knowledge transfer. Theoretical results establish convergence guarantees and alignment bounds. Extensive offline simulations and real-world A/B tests demonstrate consistent improvements over both loss-based and gradient-based multi-task baselines.

**Strengths:**

1. **Clear problem motivation with empirical grounding.** The paper provides compelling evidence of distribution shift challenges in online advertising with detailed visualizations of hourly impression volumes and value distributions across multiple days, showing clear temporal volatility and task-specific distributional changes.

2. **Strong theoretical foundation and sound derivations.** This paper provides detailed convergence analysis under well-defined smoothness and bounded-gradient assumptions. The proofs are mathematically rigorous and clearly connect the proposed algorithm to its theoretical guarantees.

3. **Comprehensive experimental validation.** The experimental design includes both controlled offline simulations and large-scale online A/B testing, showing measurable improvements across multiple key business metrics.

**Weaknesses:**

1. **Strong and idealized theoretical assumptions.** The convergence proofs rely on assumptions such as “gradient coverage” and “bounded smoothness,” which may not hold in stochastic or adversarial advertising environments. The theory assumes deterministic gradients, while the actual optimization involves noisy mini-batch estimates. This discrepancy reduces the strength of the theoretical claims for real-world applications.

2. **Limited experimental scope and baseline coverage.** Real-world experiments (Table 2) compare only against the vanilla baseline due to operational constraints, providing insufficient evidence of superiority over other sophisticated multi-task methods in production settings.

3. **Scalability and Computational Efficiency.** The method is evaluated on simulated and real-world datasets, but the scalability of the approach, especially with a large number of tasks or on larger advertising platforms, is not fully addressed. There is no detailed discussion on the computational cost or memory overhead, which is critical for real-time bidding systems.

**Questions:**

1. VAMO is designed for online auto-bidding, but the mechanism is conceptually general. Can this validation-aligned weighting generalize to other domains (e.g., recommendation or finance) with different task correlations and dynamics?

2. Are there empirical or theoretical limits to its transferability?

---

### Official Review · Reviewer_6BRx · 2025-11-01

**Soundness:** 1
**Presentation:** 2
**Contribution:** 2
**Rating:** 2
**Confidence:** 4

**Summary:**

The paper proposes VAMO—a validation‑aligned multi‑task optimization scheme for auto‑bidding—and a shared‑bottom, task‑specific generative architecture augmented with a periodicity‑aware temporal module (based on TimesNet/FFT reshaping). At each step, task weights are set by a softmax over dot products between the validation gradient and per‑task training gradients (Algorithm 1). The authors provide a convergence bound under smoothness, bounded‑gradient, and “alignment coverage” assumptions, and report improvements on a 10‑day simulation and an online A/B on Taobao (Tables 1–2)

**Strengths:**

The proposed algorithm is easy to implement and avoids hypergradient computation.

The numerical experiments are extensive and effectively demonstrate the performance of the proposed method.

**Weaknesses:**

1. My main concern is that the theoretical result (Theorem 1) may be incorrect. The Taylor expansion in Equation (24) is invalid because $ g_i^{\mathrm{val}} $ is evaluated on the minibatch $B^{\mathrm{val}} $. Therefore, $ g_i^{\mathrm{val}} $ does not equal $\nabla_\theta \mathcal{L}^{\mathrm{val}}(\theta) $.
2. Eq. (6) uses $O(|\Delta\theta|)$ where the correct remainder is $O(|\Delta\theta|^2)$.

3. Some assumptions are too strong and non standard.
   a. Assumption 1 uses
 $
   |L_{\text{val}}(\theta_i)-L_{\text{val}}(\theta_j)| \le L|\theta_i-\theta_j|^2,
   $
   which is stronger than usual Lipschitz gradient smoothness $O(|\Delta\theta|^2$ remainder). In fact, the square loss does not satisfy this assumption.
   b.  Assumption 3 requires a convex combination of task gradients to have a uniform positive inner product with the validation gradient. This is rarely satisfied in nonstationary online systems. The claim that it is “mild” is overstated.
c.    The “norm comparability” (M) in Assumption 3 never enters the proof.

4. Typos:   The simplex alternates between $\Delta_{K}$ and $\Delta_{K-1}$; unify notation.

**Questions:**

It seems that the proposed method can be applied to general multi-task learning problems. What makes it particularly suitable or unique for the auto-bidding setting?

---

### Note · Authors · 2026-01-06

I have read and agree with the venue's withdrawal policy on behalf of myself and my co-authors.